# ExoBDNF Probiotic Supplementation Enhances Cognition in Subjective Cognitive Decline

**DOI:** 10.3390/medicina62010091

**Published:** 2025-12-31

**Authors:** Ching-En Lin, Li-Fen Chen, Wen-Hui Fang, Chuan-Chia Chang, Hsin-An Chang

**Affiliations:** 1Department of Psychiatry, Taipei Tzu Chi Hospital, New Taipei City 231, Taiwan; chingenlinaa@gmail.com; 2School of Medicine, Buddhist Tzu Chi University, Hualien 970, Taiwan; 3Graduate Institute of Medical Sciences, National Defense Medical University, Taipei 114, Taiwan; annchen5133@gmail.com; 4Taoyuan Psychiatric Center, Ministry of Health and Welfare, Taoyuan 330, Taiwan; 5Department of Family and Community Medicine, Tri-Service General Hospital, National Defense Medical University, Taipei 114, Taiwan; rumaf.fang@gmail.com; 6Department of Psychiatry, Tri-Service General Hospital, National Defense Medical University, Taipei 114, Taiwan; changcc@mail.ndmctsgh.edu.tw

**Keywords:** cognition, gut microbiota, probiotics

## Abstract

*Background and Objectives*: Interventions targeting the gut–brain axis offer potential for mitigating Subjective Cognitive Decline (SCD), a critical window for Alzheimer’s prevention. This study evaluated the effects of a novel probiotic supplement, ExoBDNF, on cognitive function, sleep, and emotional distress in adults with SCD. *Materials and Methods*: In this 9-week open-label study, participants received ExoBDNF supplementation. Efficacy was assessed using the SCD-Questionnaire (SCD-Q), DASS-21, PSQI, MoCA, and a computerized cognitive battery measuring inhibition (Go/No-Go), flexibility (Task Switching), and working memory. *Results*: Post-intervention analyses revealed significant improvements in subjective cognition (SCD-Q, *p* < 0.001), sleep quality (PSQI, *p* < 0.001), and emotional distress (DASS-21, *p* < 0.001). Objective cognitive performance also improved, with significant gains in MoCA scores (*p* = 0.047) and executive function metrics. Spearman correlation analysis indicated a significant link between cognitive and emotional changes: longitudinal reductions in SCD scores correlated with concurrent reductions in emotional distress (rho = 0.471, *p* = 0.009). Furthermore, higher baseline SCD scores predicted greater improvement in emotional outcomes (rho = −0.540, *p* = 0.002). *Conclusions*: ExoBDNF supplementation significantly enhanced cognitive performance, sleep quality, and emotional well-being. The findings demonstrate that improvements in subjective cognition are closely tied to alleviated emotional distress, supporting the gut–brain axis as a viable therapeutic target for early-stage cognitive decline.

## 1. Introduction

Cognition encompasses the brain’s neural processes and signal integration mechanisms that facilitate learning, memory, and decision-making. Dementia is characterized by a decline in cognitive function severe enough to impair an individual’s ability to perform daily activities independently. The World Health Organization (WHO) estimates that over 55 million individuals globally are affected by dementia, with prevalence rates increasing annually. This trend imposes significant economic burdens worldwide and escalates caregiving demands on families. Cognitive decline generally advances through progressive stages, and timely intervention with suitable strategies may mitigate the progression of neurodegeneration [1].

Subjective cognitive decline (SCD) refers to an individual’s perception of a decline in cognitive abilities, despite performing within the normal range on objective cognitive tests. In recent years, it has been considered a possible transitional stage to mild cognitive impairment (MCI) and an early risk factor for dementia [2]. Mild cognitive impairment is often a precursor to dementia in older adults; although patients typically maintain their ability to perform daily self-care activities, memory impairments have already begun to appear. Currently, there are limited treatment options available for SCD and MCI, highlighting the need to find more effective interventions to slow cognitive decline.

Good exercise and dietary habits, as well as improved sleep quality, help prevent or treat cognitive impairment by modulating neuroinflammation, metabolic health, and the gut–brain axis, which is a bidirectional communication network involving the central nervous system (CNS), autonomic nervous system, enteric nervous system (ENS), and the hypothalamic–pituitary–adrenal (HPA) axis [3]. Gut microbiota are key regulators of the gut–brain axis, influencing host physiology and cognitive function through the production of neuroactive molecules and modulation of immune and endocrine pathways. Microbial metabolites such as short-chain fatty acids (SCFAs), gamma-aminobutyric acid (GABA), and serotonin precursors are secreted by gut bacteria and can signal to the brain via the enteric nervous system, vagus nerve, and systemic circulation, thereby affecting central nervous system activity and cognitive processes [4,5]. The brain, in turn, can influence gut microbial composition and function through the release of neurotransmitters and neuroendocrine signals [4].

Alterations in gut microbiota composition—often driven by aging, stress, or dietary habits—can increase intestinal permeability and promote local and systemic inflammation, which may contribute to neuroinflammation and cognitive decline [6,7]. Diets rich in fiber, polyphenols, and prebiotics support beneficial microbial populations, enhance SCFA production, and reduce inflammatory signaling, while Westernized diets high in fat and low in fiber promote dysbiosis and increased risk of cognitive impairment [5,7,8]. Interventions such as probiotics, prebiotics, and psychobiotics have shown potential to improve cognitive outcomes by restoring microbial balance and modulating neuroimmune and neuroendocrine pathways [9,10]. In summary, dietary habits can modulate gut microbial composition, which in turn impacts the gut–brain axis and cognitive function through neuroactive metabolites, immune signaling, and regulation of intestinal barrier integrity [5,7,8].

Growing evidence indicates a strong bidirectional relationship between the gut microbiota and brain health. The gut–brain axis operates through several key mechanisms. First, neurotransmitters and short-chain fatty acids (SCFAs) produced in the gut can directly signal the brain via the vagus nerve, whereas long-chain fatty acids activate vagal pathways through cholecystokinin (CCK) [11,12]. Second, hormones, neurotransmitters, and immune signaling molecules are able to cross the intestinal barrier, enter the systemic circulation, and ultimately reach the central nervous system through the blood–brain barrier (BBB) [13,14]. Third, microbial-associated molecular patterns (MAMPs) and microbial metabolites can stimulate peripheral immune cells, which in turn modulate neuroinflammation and brain function [15,16]. Together, these pathways highlight the potential of gut microbiota–targeted interventions to influence cognitive processes and neurological health.

Current evidence indicates that *Lactococcus lactis* is a safe, non-pathogenic, and edible strain, with genomic analyses confirming the absence of virulence genes. Studies have further shown that *L. lactis* produces substantial amounts of extracellular vesicles (EVs) that carry bioactive molecules with potential regulatory functions. In neuroprotection research, EVs and secreted metabolites derived from lactic acid bacteria, including *Lactococcus* and *Lactobacillus* species, have been reported to enhance brain-derived neurotrophic factor (BDNF) expression and mitigate amyloid beta 1–42–induced neuronal toxicity [17,18,19]. Despite growing evidence linking the gut microbiota to cognitive health through neurochemical, immune, and vagal pathways, clinical trials investigating microbiome-based interventions in early cognitive decline remain limited. *Pediococcus acidilactici*, which is taxonomically separate from the genus *Lactobacillus*, represents an additional genus within the lactic acid bacteria group [20]. A 2025 animal study revealed that oral administration of *Pediococcus acidilactici* to mice experiencing antibiotic-induced dysbiosis effectively reinstated BDNF expression within the hippocampus [21]. *Pediococcus acidilactici* metabolizes tryptophan to generate serum indole-3-lactic acid (ILA), which traverses the blood–brain barrier through systemic circulation to activate the aryl hydrocarbon receptor (AHR) signaling pathway. This activation subsequently mitigates neuroinflammation and alleviates depressive-like behaviors in murine models [22]. Although numerous animal studies have been conducted, there remains a lack of human research investigating probiotics genetically modified to modulate neurotrophic pathways—specifically ExoBDNF-producing strains such as *Pediococcus acidilactici*—and their impact on cognitive function, psychological well-being, and sleep quality in individuals experiencing Subjective Cognitive Decline (SCD) or Mild Cognitive Impairment (MCI). This lack of evidence represents a key gap in identifying feasible and non-pharmacological strategies to slow cognitive deterioration. Therefore, the aim of this study was to investigate the effects of an ExoBDNF-producing lactic acid bacteria (*Pediococcus acidilactici*) supplement on cognitive function, psychological outcomes, and sleep quality in individuals with SCD or MCI using an open-label, single-group pretest–posttest design. This pilot trial seeks to provide preliminary evidence to support the development of future randomized controlled studies targeting the gut–brain axis in early cognitive decline.

## 2. Method

### 2.1. Participants

Participants were eligible for inclusion if they were adults aged 18 years or older who were fully conscious, able to communicate, and willing to participate, and if they voluntarily provided written informed consent; a total of 30 volunteers meeting these criteria were enrolled in the study.

### 2.2. Inclusion Criteria

Participants met criteria for either Subjective Cognitive Decline (SCD) or Mild Cognitive Impairment (MCI). Individuals classified as SCD reported a persistent decline in memory over the past five years, corroborated by an informant such as a family member, friend, or colleague. SCD was further defined by a Montreal Cognitive Assessment (MoCA) score greater than 25, a Mini-Mental State Examination (MMSE) score greater than 27, and a total score of 0–24 on the Subjective Cognitive Decline Questionnaire. Individuals classified as MCI were required to have a MoCA score between 11 and 25 and an MMSE score between 24 and 27.

### 2.3. Exclusion Criteria

Participants were excluded if pregnancy was confirmed or expected, or if they had a history of gallbladder or gastrointestinal disease, gout, porphyria, gastric bariatric surgery, hypertension (≥160/100 mmHg after 10 min of rest), diuretic use, cardiac disease, hepatic or renal dysfunction, thyroid disorders, Cushing’s syndrome, malignancy, severe sensory impairment, intellectual disability, or any condition that could interfere with study outcomes.

They were also excluded if they had a history of brain surgery, severe traumatic, neurovascular, infectious, or other major brain injury, epilepsy, or neurological disorders—including traumatic brain injury with loss of consciousness longer than 24 h or post-traumatic amnesia lasting more than 7 days—or if they had consumed probiotics within two weeks prior to screening or participated in another clinical trial within four weeks before screening. In addition, participants were excluded if they had a history of severe psychiatric disorders (e.g., psychotic disorders, bipolar disorder, or severe major depressive disorder requiring hospitalization) or neurological conditions that could substantially interfere with cognitive assessment. Individuals with mild emotional symptoms, such as subclinical depressive or anxiety symptoms, were not excluded, as these are commonly observed in individuals with SCD and were evaluated as part of the study outcomes.

### 2.4. Study Procedures

This study was designed as a single-group pretest–posttest, open-label trial. At baseline (week 0), enrolled participants underwent physical examinations and completed initial questionnaires. Starting from the first week, daily administration of ExoBDNF (*Pediococcus acidilactici*) supplementation (ClinicalTrials.gov ID: NCT06968299) was initiated at a dosage of 1 × 10^10^ colony-forming units (CFU) [23], which was maintained through the end of week 9. Throughout the intervention period, participants were instructed to maintain their usual dietary and lifestyle habits.

After starting the diet, subjects filled out a daily diary that included questions about study product intake, other food intake, bowel movement frequency, stool quality (consistency and color), any medications received, and any unpleasant symptoms such as diarrhea, constipation, vomiting, flatulence, and discomfort. It is expected that this project will help clarify whether supplementation with ExoBDNF (*Pediococcus acidilactici*) can help improve cognitive ability, sleep or mental health (The ethics approval date: 14 May 2025; The trial registration date: 5 May 2025; The study start date for this experiment: 20 May 2025).

### 2.5. Outcome Measurement

The primary endpoint was defined as the change in the Subjective Cognitive Decline Questionnaire (SCD-Q) score from baseline to week 9 [24]. This comparison aimed to assess whether a statistically significant difference existed between the baseline and week 9 measurements.

The SCD-Q serves as an instrument to evaluate self-perceived cognitive deterioration by systematically documenting an individual’s subjective experience of cognitive changes over the preceding two years. The SCD-Q comprises two principal components: “MyCog”, completed by the individual, and “TheirCog”, completed by an informant or caregiver. The “MyCog” section contains 24 items designed to assess perceived alterations in memory, language, and executive functioning. Each component of the SCD-Q—both “MyCog” (self-report) and “TheirCog” (informant-report)—yields a score ranging from 0 to 24. Items are scored dichotomously (yes/no), with higher scores reflecting greater perceived cognitive decline. This scoring methodology is substantiated by validation studies documented in the medical literature [25,26].

Secondary endpoints included computerized cognitive assessments using the Cognitive Test Battery [27], comprising a Go/No-Go task to evaluate attention and response inhibition, a task-switching paradigm to assess cognitive flexibility, and a working memory task (Corsi block-tapping) to measure spatial short-term memory. Participants also completed validated questionnaires, including the Mini-Mental State Examination (MMSE) [28] and Montreal Cognitive Assessment (MoCA) [29]. The MoCA provides a comprehensive, highly sensitive assessment of attention and memory through tasks such as digit span, vigilance, serial subtraction, five-word learning, and delayed recall with cueing. It also evaluates executive and visuospatial functions, enabling the detection of subtle cognitive deficits—especially mild cognitive impairment (MCI) and early dementia—and helps distinguish amnestic from non-amnestic cognitive syndromes [30]. MMSE is more effective for detecting moderate-to-severe dementia but frequently underestimates impairment in patients with mild cognitive deficits or single-domain impairment [31].

Other secondary endpoints included the Taiwan Cognition Questionnaire (TCQ) [32], Depression Anxiety Stress Scales-21 (DASS-21) [33], and the Pittsburgh Sleep Quality Index (PSQI) [34]. The TCQ assesses subjective cognitive complaints in Taiwanese adults, focusing on memory, attention, and executive function. Culturally adapted for the population of Taiwan, it is a self-report tool useful for epidemiological research and cognitive screening. It allows quick identification of perceived cognitive difficulties that may aid early detection of cognitive impairment [35]. The DASS-21 is a short self-report measure assessing depression, anxiety, and stress through three seven-item subscales rated on a 4-point Likert scale. Higher scores indicate greater severity of emotional disturbance. It demonstrates strong structural validity and internal consistency, allowing clear distinction among the three emotional states. Widely used in clinical and research settings, it serves as an efficient tool for screening psychological distress [36,37]. The PSQI is a widely used self-report measure of sleep quality over the past month, covering seven domains: subjective sleep quality, sleep latency, sleep duration, habitual sleep efficiency, sleep disturbances, use of sleep medication, and daytime dysfunction. It yields a global score distinguishing good and poor sleepers and has been validated in clinical and non-clinical samples, including those in Taiwan. Its strengths include comprehensive coverage of sleep domains, strong reliability, and high sensitivity to sleep dysfunction [38,39].

### 2.6. Statistics

Comparative analyses between pre-intervention and post-intervention time points were performed utilizing either the paired *t*-test or the Wilcoxon signed-rank test, implemented through SPSS statistical software version 24.0 (IBM Corp., Chicago, IL, USA). Data are presented as mean ± standard deviation (SD). Normality was assessed using the Kolmogorov–Smirnov test. A *p*-value of less than 0.05 was considered statistically significant. Correlation between SCD-Q and DASS-21 data was analyzed by Spearman correlation analysis. The intervention was considered to have a supportive modulatory effect if statistically significant differences were observed in within-subject comparisons, accompanied by improvements in key clinical symptoms.

## 3. Results

A total of 30 participants, comprising 29 individuals with subjective cognitive decline (SCD) and 1 individual with mild cognitive impairment (MCI), completed the 9-week ExoBDNF supplementation protocol. The mean age of the participants was 40.97 years (±12.05), with females representing 70% of the sample. The average duration of education was 16.33 years, with a standard deviation of ±1.97 years. Only one individual was diagnosed with primary insomnia disorder and was concurrently using hypnotic medication; the other participants exhibited no diagnosed mental health disorders nor were they undergoing any significant psychotropic treatment. Table 1 summarizes baseline (week 0) and post-intervention outcomes (week 9).

### 3.1. Subjective Cognitive Outcomes

SCD-Q scores decreased from 50.6 ± 6.4 at baseline to 37.9 ± 8.4 at week 9 (t = 7.8, *p* < 0.001), indicating reduced self-perceived cognitive decline. TCQ scores improved from 7.7 ± 3.1 to 3.4 ± 2.3 (t = 7.3, *p* < 0.001), reflecting reduced subjective cognitive complaints.

### 3.2. Psychological and Sleep Measures

DASS-21 total scores decreased from 22.0 ± 12.6 to 11.6 ± 9.0 (z = −4.5, *p* < 0.001), indicating significant reductions in psychological distress. PSQI scores declined from 8.5 ± 2.3 to 6.3 ± 2.9 (t = 4.0, *p* < 0.001), suggesting better overall sleep.

### 3.3. Global Cognitive Function

MoCA scores increased from 27.5 ± 2.0 to 28.3 ± 2.3 (z = −2.0, *p* = 0.047). MMSE scores showed no significant change (29.1 ± 1.2 to 29.2 ± 1.0, *p* = 0.542).

### 3.4. Computerized Cognitive Test Battery

#### 3.4.1. Go/No-Go (Attention and Inhibitory Control)

Correct response rate improved significantly (95.0 ± 6.3% to 97.3 ± 6.0%; z = −2.0, *p* = 0.046). Reaction time showed no significant change.

#### 3.4.2. Task-Switching (Cognitive Flexibility)

Correct rate showed a trend toward improvement but did not reach significance (*p* = 0.053). Reaction time improved substantially (1063.0 ± 287.0 s to 919.3 ± 215.7 s; z = −3.8, *p* < 0.001).

#### 3.4.3. Working Memory (Corsi Block-Tapping)

Global WM performance was largely unchanged in accuracy (50.8 ± 20.1% to 52.0 ± 20.8%; *p* = 0.779), but reaction time improved modestly (3827.0 ± 908.6 s to 3542.5 ± 555.2 s; z = −2.1, *p* = 0.039).

### 3.5. Correlation Between SCD-Q and DASS-21 Data

Baseline scores on the SCD-Q demonstrated a significant negative correlation with the changes in DASS-21 scores observed between week 1 and week 9 (Spearman’s r = −0.54, *p* = 0.002).

The variations in SCD-Q scores demonstrated a positive correlation with the variations in DASS-21 scores from week 1 to week 9 (Spearman’s r = 0.471, *p* = 0.009). Figure 1 illustrates the scatter plots with best-fit regression lines and 95% confidence intervals. First, a significant negative correlation was observed between baseline SCD scores and the longitudinal change in DASS-21 scores (rho = −0.54, *p* = 0.002). This suggests that participants with higher initial subjective cognitive complaints tended to show a greater decrease in emotional distress levels over time. Second, we found a significant positive correlation between the change in SCD scores and the change in DASS-21 scores (rho = 0.471, *p* = 0.009). This indicates that the longitudinal trajectory of subjective cognitive decline is closely associated with changes in emotional states; specifically, reductions in SCD scores were associated with concurrent reductions in DASS-21 scores.

### 3.6. Safety and Tolerability

ExoBDNF supplementation was well tolerated throughout the 9-week intervention period. No adverse events potentially related to the study product were reported. Specifically, no gastrointestinal symptoms such as diarrhea, constipation, flatulence, or vomiting were observed, and no participants discontinued the intervention due to adverse effects.

## 4. Discussion

This pilot study provides initial evidence that ExoBDNF (*Pediococcus acidilactici*) supplementation may enhance cognitive function, psychological well-being, and sleep quality in individuals with early cognitive decline. Significant reductions in subjective cognitive complaints (SCD-Q, TCQ) and modest improvements in global cognition (MoCA) suggest that the intervention may benefit individuals at a stage where cognitive changes are subtle yet clinically meaningful. Because subjective decline often precedes measurable impairment, these findings highlight the potential of ExoBDNF as an early-stage supportive strategy.

Improvements in psychological distress and sleep quality further reinforce the intervention’s relevance, as mood disturbances and poor sleep are closely linked to worsening cognitive outcomes [40,41]. The concurrent changes across these domains suggest that ExoBDNF may modulate shared gut–brain pathways involved in emotional regulation, stress response, and cognitive processing. In terms of safety, ExoBDNF supplementation was well tolerated, with no treatment-related adverse events reported. This finding is consistent with prior evidence indicating that probiotic interventions are generally safe in clinical populations.

Concurrently with the present investigation, meta-analyses of randomized controlled trials have consistently revealed enhancements in global cognitive function, memory, processing speed, and spatial ability subsequent to probiotic supplementation. These improvements are characterized by standardized mean differences ranging from 0.4 to 0.6 in older adult populations and individuals experiencing cognitive impairment [42,43]. The effect is most pronounced with single-strain formulations, higher doses (e.g., ≥1 × 10^9^ CFU/g), and intervention durations of at least 12 weeks [42,43]. Mechanistically, probiotics are proposed to confer cognitive advantages through the modulation of the gut–brain axis. This includes the attenuation of neuroinflammatory processes, the upregulation of neurotrophic factors such as BDNF, and the regulation of neurotransmitter metabolism [42,44]. In healthy adults and younger populations, the evidence for cognitive enhancement is less robust and often inconsistent, with some studies showing no significant effect [45,46].

At the neural systems level, studies in schizophrenia have demonstrated that dysfunctional connectivity—particularly excessive high-frequency resting-state synchrony in regions such as the cuneus, superior temporal gyrus, and fusiform gyrus—is strongly associated with attentional impairments. These findings highlight that cognitive deficits can arise from maladaptive large-scale oscillatory patterns that impair the precision of information processing [47]. Such evidence converges with neuromodulation research showing that targeted modulation of cortical excitability (e.g., rTMS, tDCS, tACS) can selectively improve working memory, attention, and executive function through rebalancing dysfunctional prefrontal–temporal–parietal circuits [48]. Integrating these perspectives, the current findings suggest that ExoBDNF supplementation may support cognition through mechanisms conceptually similar to neuromodulation: both strategies ultimately influence neural plasticity and large-scale network efficiency, albeit through different biological routes. Whereas rTMS and tDCS act directly upon cortical excitability and connectivity, gut-derived metabolites and bacterial extracellular vesicles (EVs) may indirectly enhance neurotrophic signaling—such as BDNF pathways—thereby strengthening synaptic resilience and cognitive network function. The observed correlation between improved subjective cognition and reduced emotional distress further supports a systems-level view in which emotional regulation and cognitive performance share overlapping circuit-level substrates and may respond synergistically to interventions that restore network balance. Notably, the EEG-based evidence from schizophrenia research demonstrates that cognitive impairment emerges when connectivity patterns become either hyper-synchronous or inefficiently coordinated. This lends additional theoretical support to microbiome-based interventions: if gut–brain modulation enhances neuroplasticity, it may help prevent or attenuate maladaptive oscillatory states that precede measurable cognitive decline, particularly in early-stage populations such as SCD. From a translational standpoint, the convergence of probiotic-based modulation and neuromodulation research highlights a promising multimodal framework in which peripheral (microbiome) and central (brain stimulation) interventions could be combined to optimize cognitive outcomes.

Safety data demonstrate that probiotics are generally well tolerated and do not result in a statistically significant increase in adverse events relative to placebo [49]. However, the reliability of the current evidence remains variable, underscoring the need for further well-designed clinical trials to establish the most effective strain selection, dosage, and treatment duration, and to confirm efficacy in more heterogeneous populations [50].

The primary limitation of this study is its open-label, single-group pretest–posttest design, which lacks a placebo control and randomization. Without a comparison group, improvements may be influenced by expectancy effects, practice effects on cognitive tests, regression to the mean, or natural symptom fluctuations, rather than the ExoBDNF supplementation itself. Additionally, the small sample size (n = 30) and the predominance of participants with SCD limit statistical power and reduce generalizability to broader SCD/MCI populations. These design constraints prevent causal inference and highlight the need for larger, randomized, double-blind, placebo-controlled trials to confirm the efficacy of ExoBDNF supplementation. Another limitation is the reliance on self-report measures for key outcomes such as subjective cognition, psychological distress, and sleep quality. Although these instruments are validated, self-reported data are inherently vulnerable to response bias, mood-related influences, and participants’ expectations, which may inflate perceived improvements. Furthermore, the absence of biological markers—such as inflammatory cytokines, gut microbiota profiling, or BDNF levels—limits the ability to verify whether the observed changes reflect underlying neurobiological or gut–brain axis mechanisms.

Thirdly, a limitation of this study is the insufficient detailed control over concomitant medications. While participants adhered to stable medication regimens during the intervention period, certain drugs—especially psychotropic medications—may affect cognitive, emotional, or sleep-related outcomes and could potentially interact with gut–brain axis mechanisms. Consequently, the possibility of residual confounding effects cannot be entirely ruled out. Fourth, only a single participant satisfied the criteria for MCI, thereby constraining the interpretability of results within this subgroup. Accordingly, the current findings predominantly pertain to individuals experiencing SCD and should not be extrapolated to populations diagnosed with MCI. Fourth, psychiatric symptoms and educational attainment may influence cognitive test performance, particularly screening tools such as the MMSE, which are known to be sensitive to education level. Although individuals with severe psychiatric disorders were excluded, mild emotional symptoms were present in some participants and may have affected attention or memory performance. Additionally, the relatively high educational level of the sample may limit the generalizability of the findings to populations with lower educational attainment. Future studies should further control for psychiatric status and education level or apply education-adjusted cognitive measures. Lastly, a notable limitation of this study is the relatively young average age of the participants. While SCD is more frequently reported among individuals aged over 50, younger adults can also experience persistent subjective cognitive complaints, often linked to emotional distress, sleep disturbances, or stress-related factors. Such complaints in younger populations may reflect an early or prodromal phenotype rather than neurodegeneration associated with aging. Consequently, the current findings should be interpreted with caution and may not be directly applicable to older individuals with SCD. Future research should focus specifically on older adult cohorts to ascertain whether the effects of ExoBDNF supplementation observed here are relevant to age-related cognitive decline.

## 5. Conclusions

This pilot study suggests that ExoBDNF-producing *Pediococcus acidilactici* supplementation may improve subjective cognition, emotional well-being, sleep quality, and selected aspects of cognitive performance in individuals with early-stage cognitive decline. The observed association between reductions in subjective cognitive complaints and improvements in emotional distress highlights the close interaction between cognitive and affective symptoms at this early stage. Clinically, these findings indicate that gut–brain axis–targeted probiotic interventions may represent a feasible, low-risk, and non-pharmacological supportive strategy for individuals with SCD, a population for whom effective preventive options remain limited. Future studies should employ randomized, placebo-controlled designs with larger samples and incorporate biological markers to clarify mechanisms and determine whether ExoBDNF supplementation can delay progression to MCI or dementia.

## Figures and Tables

**Figure 1 medicina-62-00091-f001:**
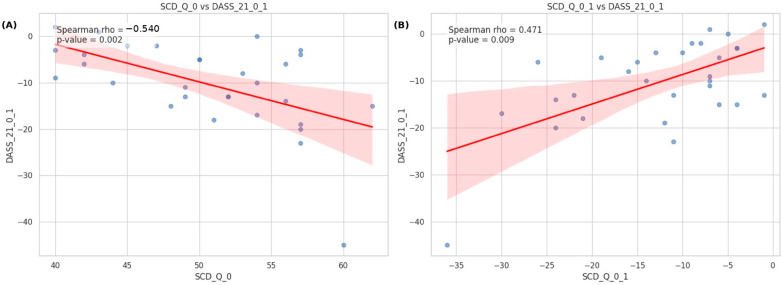
Scatter plots illustrating the associations between subjective cognitive decline (SCD) and emotional distress (DASS–21). (**A**) The relationship between baseline SCD scores and the change in DASS–21 scores. A significant negative correlation was found (rho = − 0.540, *p* = 0.002). (**B**) The relationship between the change in SCD scores and the change in DASS-21 scores. A significant positive correlation was observed (rho = 0.471, *p* = 0.009). The red solid lines represent the best-fit linear regression models, and the shaded areas indicate the 95% confidence intervals.

**Table 1 medicina-62-00091-t001:** Clinical characteristics and data collected at baseline and following the completion of ExoBDNF supplementation.

Variables	Baseline (Mean ± SD)	Week 9 (Mean ± SD)	Paired *t*-Test/ Wilcoxon Signed-Rank Test	*p*
Age	40.97 ± 12.05			
Sex (Female): N (%)	21 (70%)			
Education (years)	16.33 ± 1.97			
Questionnaire				
SCD-Q	50.6 ± 6.4	37.9 ± 8.4	t = 7.8	<0.001
TCQ	7.7 ± 3.1	3.4 ± 2.3	t = 7.3	<0.001
PSQI	8.5 ± 2.3	6.3 ± 2.9	t = 4.0	<0.001
DASS-21	22.0 ± 12.6	11.6 ± 9.0	z = −4.5	<0.001
MoCA	27.5 ± 2.0	28.3 ± 2.3	z = −2.0	0.047
MMSE	29.1 ± 1.2	29.2 ± 1.0	z = −0.6	0.542
Cognitive Test Battery				
Go/No Go				
correction rate (%)	95.0 ± 6.3	97.3 ± 6.0	z = −2.0	0.046
reaction time (s)	640.0 ± 97.1	627.5 ± 71.8	z = 0.5	0.600
Task-switching				
correction rate (%)	95.7 ± 5.2	96.4 ± 8.3	z = −1.9	0.053
reaction time (s)	1063.0 ± 287.0	919.3 ± 215.7	z = −3.8	<0.001
Working memory				
correction rate (%)	50.8 ± 20.1	52.0 ± 20.8	z = −0.3	0.779
reaction time (s)	3827.0 ± 908.6	3542.5 ± 555.2	z = −2.1	0.039

Abbreviations: SD, standard deviation; SCD-Q, Subjective Cognitive Decline Questionnaire; TCQ, Taiwan Cognition Questionnaire; PSQI, Pittsburgh Sleep Quality Index; DASS-21, Depression Anxiety Stress Scales-21items; MoCA, Montreal Cognitive Assessment; MMSE, Mini-Mental State Examination.

## Data Availability

The datasets generated during and/or analyzed during the current study are available from the corresponding author on reasonable request.

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
