# Peer review of "ExoBDNF Probiotic Supplementation Enhances Cognition in Subjective Cognitive Decline"

_medicina, 2025, doi:10.3390/medicina62010091_

Round 1

Reviewer 1 Report

Comments and Suggestions for Authors

General comments:

The paper reports on an open-label, uncontrolled pilot study of a probiotic supplement in adults with subjective cognitive decline. The methods are appropriate to explore safety and possible treatment effects and to obtain useful information for the planning of a later randomized controlled trial. The limitations of the study design are acknowledged in the Discussion and no inappropriate conclusions are drawn.

Specific issues:

  • Introduction/Methods/Discussion: A clear description of the treatment is missing. In the Introduction it is stated that the aim of the study was to investigate the effects of an ExoBDNF-producing lactic acid bacteria supplement. Under section 4. Study procedures, the daily administration of ExoBDNF supplementation at a dosage of 1 x 1010 colony-forming units is mentioned in the first paragraph, and the species Pediococcus acidilactici is mentioned. From these different pieces of information, I conclude that an ExoBDNF-producing strain of Pediococcus acidilactici was administered at a daily dose of 1 x 1010 colony-forming units. If this is correct, it should be written as one statement in section 2.4. In the first paragraph of the Discussion, it is stated that the study provides initial evidence that ExoBDNF-producing Lactococcus lactis supplementation may enhance cognitive function. Is Pediococcus acidilactici a sub-species of Lactococcus lactis? Please clarify.
  • Introduction: Some information on Pediococcus acidilactici should be provided, to justify the use of this specific bacterium for this purpose.
  • Methods, Participants: Subjective cognitive decline is more a problem of people over 50 years of age. What was the reason to include much younger subjects? The young age of the study participants should also be addressed in the Discussion.
  • Results, Table 1: For the Cognitive Test Battery, is not the correct unit for reaction time milliseconds (ms)? Please check.
  • Results: Safety and tolerability of the treatment should also be reported here. Were there any adverse events which might have been related to the treatment?

Author Response

Comment 1:

Introduction/Methods/Discussion: A clear description of the treatment is missing. In the Introduction it is stated that the aim of the study was to investigate the effects of an ExoBDNF-producing lactic acid bacteria supplement. Under section 4. Study procedures, the daily administration of ExoBDNF supplementation at a dosage of 1 x 1010 colony-forming units is mentioned in the first paragraph, and the species Pediococcus acidilactici is mentioned. From these different pieces of information, I conclude that an ExoBDNF-producing strain of Pediococcus acidilactici was administered at a daily dose of 1 x 1010 colony-forming units. If this is correct, it should be written as one statement in section 2.4. In the first paragraph of the Discussion, it is stated that the study provides initial evidence that ExoBDNF-producing Lactococcus lactis supplementation may enhance cognitive function. Is Pediococcus acidilactici a sub-species of Lactococcus lactis? Please clarify.

Response 1: Thank you for your reminder. I have clarified that the species Pediococcus acidilactici is not a subspecies of Lactococcus lactis, and I have made efforts to more clearly identify the study species, as indicated on page 6, line 15.

Pediococcus acidilactici, which is taxonomically separate from the genus Lactobacillus, represents an additional genus within the lactic acid bacteria group [20]. A 2025 animal study revealed that oral administration of Pediococcus acidilactici to mice experiencing antibiotic-induced dysbiosis effectively reinstated BDNF expression within the hippocampus [21]. Pediococcus acidilactici metabolizes tryptophan to generate serum indole-3-lactic acid (ILA), which traverses the blood-brain barrier through systemic circulation to activate the aryl hydrocarbon receptor (AHR) signaling pathway. This activation subsequently mitigates neuroinflammation and alleviates depressive-like behaviors in murine models [22].

Comment 2:

Introduction: Some information on Pediococcus acidilactici should be provided, to justify the use of this specific bacterium for this purpose.

Response 2: Thank you for your reminder. In response to comment 1, relevant information concerning Pediococcus acidilactici has been included, as indicated on page 6, line 15.

Comment 3:

Methods, Participants: Subjective cognitive decline is more a problem of people over 50 years of age. What was the reason to include much younger subjects? The young age of the study participants should also be addressed in the Discussion.

Response 3: We thank the reviewer for this important comment. Although subjective cognitive decline (SCD) is more prevalent in older adults, SCD is defined by subjective cognitive complaints rather than chronological age, and younger adults may also report persistent cognitive concerns, particularly in the context of stress, sleep disturbance, or affective symptoms.

In the present pilot study, younger participants were included to explore the potential effects of ExoBDNF supplementation across a broader SCD spectrum and to enhance feasibility in early-stage or preclinical populations. This approach aligns with emerging evidence suggesting that subjective cognitive complaints in midlife may represent an early risk phenotype rather than a benign phenomenon.

We agree that the relatively young mean age of our sample is a limitation and have now explicitly addressed this issue in the Discussion section, as indicated on page 18, line 21. We have also clarified that the findings should be interpreted cautiously and may not be directly generalizable to older SCD populations. Future randomized controlled trials focusing on individuals aged ≥ 50 years are warranted to confirm clinical applicability.

Lastly, a notable limitation of this study is the relatively young average age of the participants. While subjective cognitive decline (SCD) is more frequently reported among individuals aged over 50, younger adults can also experience persistent subjective cognitive complaints, often linked to emotional distress, sleep disturbances, or stress-related factors. Such complaints in younger populations may reflect an early or prodromal phenotype rather than neurodegeneration associated with aging. Consequently, the current findings should be interpreted with caution and may not be directly applicable to older individuals with SCD. Future research should focus specifically on older adult cohorts to ascertain whether the effects of ExoBDNF supplementation observed here are relevant to age-related cognitive decline.”

Comment 4:

Results, Table 1: For the Cognitive Test Battery, is not the correct unit for reaction time milliseconds (ms)? Please check.

Results: Safety and tolerability of the treatment should also be reported here. Were there any adverse events which might have been related to the treatment?

Response 4: Thank you for your reminder. I have checked and revised it to milliseconds (ms) in accordance with the updated table 1. We also thank the reviewer for this important comment. Safety and tolerability were monitored throughout the study using daily participant diaries, which recorded gastrointestinal symptoms and other discomforts. No adverse events potentially related to ExoBDNF supplementation—including diarrhea, constipation, flatulence, vomiting, or other clinically relevant symptoms—were reported during the 9-week intervention period, as indicated on page 13, line 20 and page 15, line 17.

3.6 Safety and Tolerability

ExoBDNF supplementation was well tolerated throughout the 9-week intervention period. No adverse events potentially related to the study product were reported. Specifically, no gastrointestinal symptoms such as diarrhea, constipation, flatulence, or vomiting were observed, and no participants discontinued the intervention due to adverse effects.”

“In terms of safety, ExoBDNF supplementation was well tolerated, with no treatment-related adverse events reported. This finding is consistent with prior evidence indicating that probiotic interventions are generally safe in clinical populations.”

Reviewer 2 Report

Comments and Suggestions for Authors

The topic is very interesting and very useful regarding the clinical implications. The authors cited a lot of recent references useful to support the clinical research.

Regarding material and method the exclusion criteria should specify if all of these participants have an actual or previous mental disorder (like depression or anxiety disorder) because some psychiatric symptoms can influence the evaluation of the cognitive function(attention, memory). Also, the level of education of the participants need to be mentioned because this issue can influence the results of the cognitive evaluation, for example MMSE, which is a screening tool and can be influenced by the level of education.

The tables and figures reflect the results of the study, but the authors ca insert a table with general information of the patient (age, level of education, relevant medical and psychiatric history, concomitant medication, etc)

It is important to mention and discuss about any concomitant medication because they can have interactions with study drug. Also, only 1 participant had a MCI maybe it is better to have a separate discussion about this case  (like a case report).

At conclusion the authors can detailed the eventually future project and the impact of the actual research on clinical cases.

The English language is understandable and correct.

Author Response

Comment 1:

Regarding material and method, the exclusion criteria should specify if all these participants have an actual or previous mental disorder (like depression or anxiety disorder) because some psychiatric symptoms can influence the evaluation of the cognitive function (attention, memory). Also, the level of education of the participants needs to be mentioned because this issue can influence the results of the cognitive evaluation, for example MMSE, which is a screening tool and can be influenced by the level of education.

Response 1: We thank the reviewer for this important methodological comment. Participants with major psychiatric disorders that could substantially interfere with cognitive assessment (e.g., psychotic disorders, severe neurological conditions) were excluded. Mild psychiatric symptoms were not used as exclusion criteria, as emotional distress is commonly observed in individuals with subjective cognitive decline and was one of the clinical dimensions evaluated in this study.

We have now clarified the exclusion criteria in the Methods (refer to page 8, line 16) section to specify psychiatric conditions more explicitly. In addition, we have added information regarding participants’ educational level, which is now reported in the Results section, as indicated on page 12, line 2.

In addition, participants were excluded if they had a history of severe psychiatric disorders (e.g., psychotic disorders, bipolar disorder, or severe major depressive disorder requiring hospitalization) or neurological conditions that could substantially interfere with cognitive assessment. Individuals with mild emotional symptoms, such as subclinical depressive or anxiety symptoms, were not excluded, as these are commonly observed in individuals with SCD and were evaluated as part of the study outcomes.”

The average duration of education was 16.33 years, with a standard deviation of ±1.97 years. Only one participant was diagnosed with primary insomnia disorder, while the remaining participants did not present with any mental health disorders.”

We also acknowledge in the Discussion that both psychiatric symptoms and educational attainment may influence cognitive test performance, particularly screening instruments such as the MMSE, and that these factors should be carefully controlled in future randomized controlled trials, as indicated on page 18, line 14.

Fourth, psychiatric symptoms and educational attainment may influence cognitive test performance, particularly screening tools such as the MMSE, which are known to be sensitive to education level. Although individuals with severe psychiatric disorders were excluded, mild emotional symptoms were present in some participants and may have affected attention or memory performance. Additionally, the relatively high educational level of the sample may limit the generalizability of the findings to populations with lower educational attainment. Future studies should further control for psychiatric status and education level or apply education-adjusted cognitive measures.”

Comment 2:

The tables and figures reflect the results of the study, but the authors can insert a table with general information of the patient (age, level of education, relevant medical and psychiatric history, concomitant medication, etc)

Response 2: Thank you for your reminder. We have included information regarding education level (see revised Table 1), prior mental disorders, concomitant medications, and related factors, as shown on page 12, line 2.

The average duration of education was 16.33 years, with a standard deviation of ±1.97 years. Only one individual was diagnosed with primary insomnia disorder and was concurrently using hypnotic medication; the other participants exhibited no diagnosed mental health disorders nor were they undergoing any significant psychotropic treatment.

Comment 3:

It is important to mention and discuss about any concomitant medication because they can have interactions with study drug. Also, only 1 participant had MCI maybe it is better to have a separate discussion about this case (like a case report).

Response 3: We thank the reviewer for highlighting the importance of concomitant medications. In this pilot study, participants were allowed to continue their usual medications, and no changes in medication regimens were made during the intervention period. However, detailed stratified analyses based on concomitant medication use were not performed due to the small sample size.

We acknowledge that concomitant medications may potentially interact with gut–brain axis mechanisms or influence cognitive, emotional, or sleep outcomes. This limitation has now been explicitly addressed in the Discussion section, as indicated on page 18, line 6. Future randomized controlled trials will include stricter control or stratification of concomitant medications to better delineate treatment effects.

We agree with the reviewer that the inclusion of only one participant with mild cognitive impairment (MCI) limits meaningful subgroup analysis. This participant was retained for feasibility and exploratory purposes in this pilot study; however, no separate statistical or inferential conclusions were drawn for the MCI subgroup. We have clarified in the manuscript that the primary findings predominantly reflect outcomes in individuals with subjective cognitive decline (SCD). Given that a single case does not allow systematic interpretation, we did not frame this observation as a formal case report but acknowledge that future studies focusing specifically on MCI populations are warranted.

Thirdly, a limitation of this study is the insufficient detailed control over concomitant medications. While participants adhered to stable medication regimens during the intervention period, certain drugs—especially psychotropic medications—may affect cognitive, emotional, or sleep-related outcomes and could potentially interact with gut–brain axis mechanisms. Consequently, the possibility of residual confounding effects cannot be entirely ruled out.”

Comment 4:

At conclusion the authors can detailed the eventually future project and the impact of the actual research on clinical cases.

Response 4: Thank you for your suggestion. I have revised the conclusion to include potential future projects and the implications of the current research for clinical applications, as detailed on page 19, line 8.

This pilot study suggests that ExoBDNF-producing Pediococcus acidilactici supplementation may improve subjective cognition, emotional well-being, sleep quality, and selected aspects of cognitive performance in individuals with early-stage cognitive decline. The observed association between reductions in subjective cognitive complaints and improvements in emotional distress highlights the close interaction between cognitive and affective symptoms at this early stage. Clinically, these findings indicate that gut–brain axis–targeted probiotic interventions may represent a feasible, low-risk, and non-pharmacological supportive strategy for individuals with SCD, a population for whom effective preventive options remain limited. Future studies should employ randomized, placebo-controlled designs with larger samples and incorporate biological markers to clarify mechanisms and determine whether ExoBDNF supplementation can delay progression to MCI or dementia.”